# How does Quorum Sensing of Intestinal Bacteria Affect Our Health and Mental Status?

**DOI:** 10.3390/microorganisms10101969

**Published:** 2022-10-05

**Authors:** Leon M. T. Dicks

**Affiliations:** Department of Microbiology, Stellenbosch University, Private Bag X1, Matieland, Stellenbosch 7602, South Africa; lmtd@sun.ac.za

**Keywords:** quorum sensing, intestinal bacteria, health

## Abstract

The human gut is host to almost 3000 microbial species, of which 90% are bacteria. Quorum sensing (QS) molecules generated by intestinal bacteria are important in establishing species- and strain-level structures within the gut microbiome but are also used to communicate with the host. Although we do not know which QS molecules have the most direct interaction with intestinal and sensory neurons, it is clear they affect our physiological and mental health. Signals produced by bacteria are diverse and include autoinducers (AIs), homoserine lactones (HSLs), quinolines, peptides, toxins and proteases. These signaling molecules activate specific receptors in the bacterial cell wall and trigger sensors in the cytoplasm that regulate gene expressions. A better understanding of the gene structures encoding the production of QS molecules is of importance when selecting strains with neurogenerative and other probiotic properties. Furthermore, QS molecules may be used as biomarkers in the diagnosis of inflammable bowel disease (IBD), irritable bowel syndrome (IBS) and colorectal cancer (CRC). In the future, it should be possible to use QS biomarkers to diagnose neurological and psychiatric diseases such as anxiety and depression, major depressive disorder (MDD), schizophrenia, bipolar disorder, autism and obsessive-compulsive disorder (OCD).

## 1. Introduction

In a highly competitive and ever-changing environment such as the gastrointestinal tract (GIT), microbiota have developed unique methods to communicate with each other. Quorum sensing (QS) molecules produced by the gut microbiota regulate a variety of cell functions such as expression of virulence genes, formation of biofilms, competence and sporulation and usually only initiate these processes when cell numbers reach a certain density [1,2,3]. These signals include autoinducers (AIs) such as AI-1, AI-2 and AI-3; 3-oxo-C12-homoserine lactone (3-oxo-C12-HSL); C4-homoserine lactone (C4-HSL); 2-heptyl-3-hydroxy-4(1H)-quinoline; 2-heptyl-4-hydroxyquinoline (HHQ); QS peptides (QSPs); peptide pheromone Agr; pore-forming toxins such as hemolysins, leucocidins and phenol-soluble modulins (psms) and proteases. Epinephrine (Epi), norepinephrine (NE), autoinducer 3 (AI-3), fucose, ethanolamine (EA) and vitamin B12 activate specific receptors in the bacterial cell wall and trigger sensors in the cytoplasm that regulate gene expressions. Conversation between microorganisms varies and ranges from interspecies communication, self-talk or intraspecies communication to cells from one genus responding to signals generated by another genus. Cells that are unable to produce their own communication signals are “listening” to signals generated by other cells, a phenomenon referred to as “eavesdropping” [4]. The gut microbiota uses certain metabolites as QS molecules to communicate with intestinal epithelial cells (IECs). *Staphylococcus aureus*, for instance, secretes a variety of virulence factors that manipulate the immune system of the host to safeguard its own survival [5]. The effect of these survival strategies on the host often manifests in the form of immune misfunctioning, neurological disorders, diarrhea and vast changes in the gut microbiome [6].

Microbial QS could be seen as a partnership or agreement amongst microbiota and, in the GIT, between the gut microbiota and the host. This requires microbiota and the host to develop specific adaptation strategies. In a complex environment such as the GIT with close to 3000 bacterial species, strains are in constant survival mode and produce an arsenal of inorganic and organic molecules participating in inter-microbial and inter-host communications. This review addresses changes brought about by QS amongst intestinal bacteria and the gut wall. The impact these communications may have on the central nervous system (CNS) and mental health is summarized.

## 2. Interbacterial Communication

### 2.1. Gram-Negative Bacteria

Gram-negative bacteria use small molecules as autoinducers (AIs) that either target transcription factors or transmembrane two-component histidine sensor kinases [7]. Amongst these, N-acyl-homoserine lactone (AHL), a small neutral lipid molecule with a homoserine lactone (HSL) moiety linked to a 4 to 18 carbon acyl side chain [8], is the best studied. AHL is synthesized from S-adenosylmethionine (SAM), catalyzed by either LuxI or LuxM synthetases (Figure 1, left image) [9]. Not all AHLs are the same. The length of the acyl side chain and substituents at the third position of the acyl chain differ and allow a LuxR-type receptor to discriminate between signals [10]. Furthermore, some species have a single AHL synthase enzyme and produce predominantly one type of AHL, whereas others have multiple AHL synthases and produce several forms of AHL [11]. The level at which AHLs are produced depends on the availability of substrates and is tightly controlled [11]. Bacteria lacking the LuxI-type synthase have orphan or “solo” LuxR-type receptors that respond to AHLs produced by other bacterial species in the same environment. SdiA (a LuxR homolog) in *Escherichia coli* and QscR in *Pseudomonas aeruginosa* are examples of such orphan LuxR-type receptors (Figure 1). These receptors are highly conserved with a 67–84% sequence identity [12] and are also found in *Enterobacter*, *Citrobacter*, *Cronobacter*, *Klebsiella*, *Salmonella* and *Shigella* [13].

Five QS systems have been described for pathogenic *E. coli*, i.e., AI-2 signaling produced by the enzyme LuxS, SdiA signaling that suppresses cell division, AI-3/Epi/NE signaling involved in host–bacteria communication, indole signaling and extracellular death factor (EDF) signaling that triggers the activation of toxin–antitoxin systems [14]. SdiA of *E. coli* (SdiAEC) is activated by AHLs produced by *P. aeruginosa* [15]. The SdiAEC/AHL complex increases the transcription of genes in the gad operon (*gadW*, *gadE*, *yhiD* and *hdeA*) of *E. coli* [16] that encodes an acid resistance system critical to the survival of enterohemorrhagic *E. coli* (EHEC) in a low-pH environment [17].

*E. coli* uses QS to regulate virulence genes, biofilm formation, mobility, the type III secretion system (T3SS), toxicity and the production of curli [18]. QS systems in *Salmonella* regulate the pathogenicity island SPI-1 (invasion), the expression of genes encoding flagella formation and the pefI-srgC plasmid operon regulating the genes *rck* (resistance to complement killing) and *srgE* (sdiA-regulated gene E) involved in the Zipper invasion mechanism [19,20,21].

*Pseudomonas aeruginosa* has three major QS systems, namely *las*, *rhl* and *pqs* (Figure 1, right image), involved in cell-to-cell communication, control over synthesis and secretion of virulence factors, bioluminescence, biofilm formation, etc. (listed in Figure 1). The *las* system regulates both the *rhl* and *pqs* systems by initiating the expression of the AI receptors RhlR and PqsR. These receptors also act as transcriptional activators when bound to their respective AIs. Freely diffusible AHL communicates with other bacterial cells or binds to LuxR-type receptors in the cytoplasm of producing cells to form stable LuxR–AHL complexes [9], as depicted in Figure 1. These LuxR–AHL complexes bind to the *Lux box* (*las* system, Figure 1) and regulate the expression of QS genes [22,23,24]. Mutants without the *lasR* gene (Δ*lasR*) are more motile, survive stationary growth much better and produce higher levels of β-lactamase and pyocyanin [25,26]. However, Δ*lasR* mutants produce fewer exoenzymes and elicit a higher immune response in host cells, as evident from an increase in the secretion of pro-inflammatory cytokines and neutrophil recruitment [27]. Complexes similar to the LasR/LasI system, e.g., RhlR/RhlI (*rhl* system, Figure 1) described for *P. aeruginosa*, produce and detect C4-homoserine lactone (C4-HSL) [28,29]. In the absence of LuxI synthases, LuxR proteins may bind to AHLs produced by other bacterial species and initiate interspecies communication [30,31,32]. The *pqs* system (Figure 1) employs two signal molecules, i.e., 2-heptyl-3-hydroxy-4(1H)-quinoline (also referred to as *Pseudomonas* quinolone signal or PQS) and its biosynthetic precursor 2-heptyl-4-hydroxyquinoline (HHQ).

### 2.2. Gram-Positive Bacteria

Gram-positive bacteria communicate using small linear or cyclized oligopeptides (QS peptides, QSPs) consisting of 5 to 17 amino acids [2,28,33,34]. The most studied QS systems are those produced by *Bacillus*, e.g., the competence sporulation factor (CSF), a pentapeptide, and the heptapeptide SDLPFEH (PapRIV). The heptapeptide forms after cleaving of the inactive 48-amino-acid pre-peptide by NprB protease [35,36].

QSPs are extracellularly secreted with the assistance of ATP-binding cassette transporters located in the cell membrane (Figure 2) and interact with either membrane-located receptors or cytoplasmic sensors such as the proteins Rap, NprR, PlcR and PrgX [33]. In the case of *Staphylococcus aureus*, the accessory gene regulator (*agr*), a four-gene operon encoding the peptide pheromone Agr, serves as the membrane-bound sensor. Agr regulates the expression of several genes, including virulence factors such as formylated peptides, proteases and pore-forming toxins (PFTs) such as hemolysins, leucocidins and phenol-soluble modulins (psms) [37]. Strains of *S. aureus* lacking the *agr* gene (Δ*agr*) form biofilms and are more prone to causing chronic infections and bacteremia [38,39,40]. *Enterococcus faecalis* uses the Fsr-QS system, which is controlled by the four-gene locus *fsrABDC* [41]. Once cleaved, the activated peptide is intracellularly transported using a transmembrane kinase (Figure 2). A cascade of phosphorylation reactions excites the peptide and induces the expression of target genes (Figure 2). For more information on the chemical characteristics and microbial background of QSPs, the reader is referred to the Quorumpeps^®^ database [42].

## 3. Interspecies Communication

Autoinducer-2 (AI-2), a furanosyl borate diester produced by Gram-negative and Gram-positive bacteria [43], plays a key role in interspecies communication and the altering of specific behavior such as virulence, luminescence and biofilm formation [44,45,46,47]. The AI-2 system is also used by the gut microbiota to overcome stressful conditions in the GIT [48,49]. Production of AI-2 is regulated by the *luxS* gene (Figure 3). S-adenosylhomocysteine (SAH) is converted to homocysteine by SAH hydrolase (SahH) in a one-step reaction but may also be produced from the cleavage of the thioether linkage of S-ribosylhomocysteine (SRH). This is a two-step reaction that requires SAH nucleosidase (Pfs) and LuxS. The intermediate, 4,5 dihydroxy-2,3-pentanedione (DPD), is rearranged to form AI-2 (Figure 3) [50].

Genes encoding homologues of luxS have been detected in more than a third of bacterial genomes, including *Escherichia coli*, *Enterococcus faecalis*, *Campylobacter jejuni*, *S. aureus*, *Clostridium difficile*, *Bacillus* spp., *Streptococcus* spp., *Shigella flexneri*, *Helicobacter pylori*, *Salmonella enterica* serotype Typhimurium, *S. enterica* serotype Typhi [43,51,52], *Bifidobacterium* [53], *Lactobacillus* [54,55], *Eubacterium*, *Roseburia* and *Ruminococcus* [56]. Strains of *E. coli* [57], *Streptococcus pneumoniae* [58], *Streptococcus mutans* [59,60] and *Lactobacillus* [61,62] use the luxS system to regulate genes encoding bacteriocin production. The same group of signaling factors is also used by *Bifidobacterium* to combat *Salmonella* infections [63]. An engineered strain of *E. coli* with increased production of AI-2 led to the reinstatement of streptomycin-repressed Firmicutes and suppressed the growth of Bacteroidetes [64]. It can be deduced from these findings that AI-2 may be used to restore balance in the gut microbiota after antibiotic treatment. Should this strategy be followed, it will have to be carefully controlled, as cytoplasmic levels of AI-2 are regulated by LsrK kinase (Figure 3). Mutants of *E. coli* with an inactive LsrK kinase were unable to phosphorylate AI-2 and lost their QS communication abilities [47,65]. However, when co-cultured with an AI-2-producing strain of *Vibrio harveyi*, the *E. coli* mutant responded to its own AI-2 and to that produced by *V. harveyi*. An increase in the adherence of *Actinobacillus pleuropneumoniae* to epithelial cells and an increase in the expression of motility genes in *E. coli* were observed in the presence of AI-2 [66,67]. In the case of *Helicobacter pylori*, AI-2 acted as a chemorepellent and prevented biofilm formation [68]. AI-2 may have the same anti-biofilm forming effect on Bacteroidetes, which would explain the decline in cell numbers as Firmicute numbers increase.

Changes in the populations of Firmicutes and Bacteroidetes alter the level and composition of SCFAs, which in turn affect gene expressions, cytokine secretion and regulatory T cell induction [69]. All these changes influence inflammatory responses. Increased levels of AI-2 could thus restore the balance between Firmicutes and Bacteroidetes and prevent, or revert, dysbiosis, IBD, obesity, autism and stress-related disorders. However positive this seems, the idea must be considered with care, as elevated levels of AI-2 may upregulate virulence, as shown with an increase in the release of bacteriophages from *Enterococcus faecalis* and an increase in gene transfer [70]. In mice, the administration of AI-2 had no effect on the expression of cytokines but aggravated *P. aeruginosa* lung infection by interfering with QS molecules produced by the pathogen [71]. These findings clearly show that implementing AI-2 aimed at maintaining gut homeostasis is far more complex and warrants further research.

Two classes of AI-2 receptors have been identified, namely LuxP, common among members of Vibrionales, and LsrB (Figure 3), widely distributed across Proteobacteria, *Bacillus cereus* and *Bacillus anthracis* [72,73,74]. The two receptors are structurally different and share a sequence similarity of only 11% [75]. Other members of Firmicutes, including gut microbiota, may respond to AI-2 using receptors similar to LuxP and LsrB. *Streptococcus mutans* and *Staphylococcus epidermidis*, however, respond to AI-2 in the absence of these receptors [43]. The LuxS/AI-2 QS system regulates the expression of several genes, including drug resistance [50]. The effect of AI-2 on the host’s immune system is less understood. Elevated levels of AI-2 were detected in tumors associated with colorectal cancer (CRC) [76]. This correlated with an increase in the expression of genes encoding TNFSF9 (tumor necrosis factor ligand superfamily member 9), as noted in tumor-associated macrophages [76]. AI-2 could thus be an important marker for CRC and warrants more research.

## 4. Interkingdom Communication

The autoinducer-3 (AI-3)/Epi/NE interkingdom signaling system [77] promotes the expression of virulence genes in pathogens such as *S. typhimurium*, *Citrobacter rodentium* and EHEC [78]. AI-3 controls the genes encoding the attachment of EHEC to epithelial cells in the colon (Figure 4), a process leading to the destruction of microvilli and the rearrangement of the cytoskeleton to form protective pedestal-like structures [79]. The direct effect of AI-3 on humans is unknown, apart from an increase in IL-8 production by THP-1 monocytes [80]. Epi and NE recognize the AI-3 receptor (Figure 4) [79] but do not activate or modulate adrenergic signaling [80]. Further research on microbial endocrinology is required to understand the effect of variations in AI-3 levels on the host.

Intestinal pathogens such as EHEC, *Salmonella typhimurium* and *Citrobacter rodentium* regulate virulence by using a two-component QS system (TCS). In the case of EHEC, the TCS consists of the quorum-sensing *E. coli* regulators QseBC and QseEF (Figure 4). QseB is a response regulator with a receiver domain and a helix-turn-helix (HTH) DNA binding domain, QseC functions as a bacterial adrenergic receptor, QseE is a sensor kinase and QseF is a response regulator [81]. Cells sensing environmental signaling compounds such as Epi, NE, AI-3, fucose and ethanolamine (EA) respond by activating transmembrane histidine kinase receptors (shown as blue and red rectangles in Figure 4). Response regulators either activate or repress the TCS. The QseC histidine sensor activates QseB, which regulates the expression of flagella and, at the same time, represses the expression of *fusK/-R*, encoding fucose metabolism and the expression of virulence genes. QseC can also phosphorylate the response regulator KdpE, which, together with Cra, stimulates genes in the LEE operon to encode the formation of microscopic “needles” through which proteins are injected into the host cell. At the same time, microvilli on the surface of epithelial cells are eradicated, and lesions with actin-rich pedestals form, onto which EHEC cells attach (not shown in Figure 4). QseC may also activate the regulator QseF, which stimulates the production of Shiga toxin. FusK is activated by fucose and phosphorylates FusR to inhibit LEE expression.

The 3-oxo-C12-HSL produced by *P. aeruginosa* (Figure 5) is actively transported across epithelial and immune cells [2,82] and destroys the permeability of the gut wall by repressing the expression of genes encoding tight junction proteins (TJs). This leads to the re-arranging (misplacing) of occludin, tricellulin, ZO-1, ZO-3, JAM-A, E-cadherin and β-catenin and prevents mucin production [2,83,84]. This not only exposes epithelial cells to infection but also activates the mucosal immune system, leading to an increase in leucocytes and the accumulation of pro-inflammatory cytokines [85]. Furthermore, 3-oxo-C12-HSL also inhibits tumor necrosis factor (TNF)-α and IL-12 production, causes malfunctioning of the T helper cell-1 (Th1) response and stimulates Th2 to produce immunoglobulin G1 [86]. Inhibition of Th1 and Th2 T lymphocyte differentiation increases cytokine production [87], intensifies oxidative stress, stimulates apoptosis and inactivates mitochondria [2].

A structurally similar form of 3-oxo-C12-HSL, 3-oxo-C12:2-HSL, has the opposite effect on the gut wall. Instead of destabilizing epithelial cells, 3-oxo-C12:2-HSL protects the tight junction proteins occludin and tricellulin and cytoplasmic ZO-1 from pro-inflammatory cytokines such as interferon-gamma (IFN-γ), TNF-α and IL-8 [88,89,90,91]. Apart from a few pioneering studies, the impact of 3-oxo-C12:2-HSL on immune cells in the human intestinal tract remains largely unknown. Landman et al. [88] reported much lower concentrations of 3-oxo-C12:2-HSL in patients diagnosed with IBD. This suggests that 3-oxo-C12:2 HSL plays a major role in the protection of epithelial cells exposed to an immune onslaught. Further research is required to determine if 3-oxo-C12:2-HSL could be used in the treatment of IBD. This also requires a better understanding of the processes involved in 3-oxo-C12:2-HSL quorum quenching, the cleaving of AHL and the hydrolysis of the homoserine lactone (HSL) ring. Thus far, three paraoxonases (PON1, PON2 and PON3) involved in the hydrolysis of the HSL ring have been identified in the GIT of humans and other mammals [92]. Of these, PON2 is the most active [93] and is predominantly expressed in the jejunum [94]. PON1 and PON3 are expressed at lower levels in patients diagnosed with Crohn’s disease and ulcerative colitis [95]. It is thus possible that these gastrointestinal disorders may be reversed by reinstating PON1 and PON3 levels. An in-depth study on the role paraoxonases play in different areas of the GIT, and their possible application in the treatment of gastric disorders, is required.

Other examples of intestinal receptors interacting with bacterial cells or bacterial compounds are pregnane X receptors (PXR1 and PXR2) and peroxisome proliferator-activated receptors (PPARα, PPARβ/δ and PPARγ). Pregnane X receptors (PXRs) are primarily expressed by intestinal epithelial cells, but also to a lesser extent by kidney cells, T cells, macrophages and dendritic cells [96], and regulate the expression of proteins involved in detoxification and the metabolism of glucose, lipid, cholesterol and bile acid [97]. Peroxisome proliferator-activated receptors (PPARs) are widely expressed in human cells, including intestinal cells [98], and regulate energy production, lipid metabolism and inflammation. PPARα represses nuclear factor kappa B (NF-κB) signaling, which decreases the production of inflammatory cytokines [99,100]. PPARγ inhibits the activation of macrophages and the production of inflammatory cytokines such as tumor necrosis factor-alpha (TNF-α), interleukin (IL)-6 and IL-1β. These anti-inflammatory responses may restore gut dysbiosis and alleviate IBDs such as ulcerative colitis and Crohn’s disease [101].

PQS and HHQ produced by *Pseudomonas* (Figure 1) interact with lymphoid cells, dendritic cells and macrophages, leading to the suppression of innate and adaptive immune responses [102,103]. In response, the aryl hydrocarbon receptor (AhR) senses the PQS signals and alerts the immune system to activate the most beneficial immune response [104,105]. This involves the expression of IL-22 and IL-17 [106]. Activation of AhR also stimulates the p38 pathway, which initiates the apoptosis of epithelial cells [107]. This is an excellent example of “eavesdropping” on an interkingdom level.

CSF, produced by *Bacillus subtilis*, binds to the cation transporter OCTN2 (Figure 5). This activates HSP-27, mediates the uptake of organic cations and carnitine and promotes intestinal barrier integrity [108]. Once in the cell, CSF acts as a reporter monitoring changes in the behavior or composition of the gut microbiota [109]. HSP-27 acts as a protein chaperone and an antioxidant but also facilitates the refolding of damaged proteins, thus preventing apoptosis and actin cytoskeletal remodeling [2,110]. The autoinducers AI-2 and AI-3 promote the expression of the immune mediators TNSF9 and IL-8 (Figure 5), and 3-oxo-C12:2-HSL reduces the production of IL-8 by epithelial cells. Host cells retaliate against QS by the binding of Epi and NE to AI-3 receptors (Figure 5). Epithelial cells then secrete AI-2 and PON that degrade HSLs (Figure 5).

PapRIV, produced by *Bacillus*, crosses the gastrointestinal tract (GIT), albeit slowly, and enters the circulatory system from where most peptides (87%) cross the BBB in a one-directional way. It can be deduced from in vitro studies that PapRIV activates microglia and may thus play a role in gut–brain interactions [111]. According to Yorick Janssens et al. [111], the second (aspartic acid) and the fourth (proline) amino acids play a key role in the activation of microglia. PapRIV also induces the production of the pro-inflammatory cytokines IL-6 and TNFα, increases intracellular ROS and stimulates an increase in ameboid cells [112]. Autoinducer peptides (AIPs) produced by *Clostridium acetobutylicum* cross the BBB much more easily than AIPs produced by *Streptococcus pneumonia* [113]. AIPs produced by Gram-positive bacteria crossing the gut wall have been shown with in vivo studies on Caco-2 cells [102,103]. De Spiegeleer et al. [114] have shown that AIPs produced by *Staphylococcus*, *Streptococcus*, *Lactobacillus* and *Bacillus* in the GIT have pro- and anti-inflammatory effects on muscle cells. The crossing of these barriers seems to depend on the structure and size of the peptide. Small diffusible molecules produced during the degradation of signal peptides, referred to as diffusible signal factors (DSFs), may also act as autoinducers [115,116].

Signals generated by gut bacteria are recorded by specialized cells in the gut wall (Figure 6), resulting in temporary or long-lasting changes in physical or mental health. These cells differentiate between signals produced by autochthonous (endemic) and foreign, potentially pathogenic, microbiota by using pattern recognition receptors (PRRs). The pro-inflammatory properties of AHLs, associated with an increase in neutrophil activity and the differentiation of fibroblasts into myofibroblasts, are crucial for tissue regeneration [117,118]. These onslaughts on the immune system are driven by systems independent of pathogen-associated molecular pattern (PAMP) recognition pathways, TLRs and the nucleotide-binding oligomerization domain proteins Nod1 and Nod2 [99]. Toll-like receptors (TLRs) and Nod-like receptors (NLRs) specialize in the recognition of microbial cell wall components, and G protein-coupled receptors (GPRs) activate G proteins involved in hormonal regulation [119,120] (Figure 6). GPRs are localized on the surface of intestinal immune cells, leukocytes, regulatory T cells, monocytes, macrophages and colonic epithelial and mesenchymal cells, thus playing a role in anti-inflammatory and pro-inflammatory processes and barrier control [121] (Figure 6).

AhRs regulate immune responses and pathogenesis (Figure 7) [122]. Large quantities of AhR are expressed by intestinal epithelial cells and immune cells such as innate lymphoid cells, intraepithelial lymphocytes, TH17 cells and Treg cells but are also present in the liver, lung, bladder and placenta [123,124,125]. One of the key functions of AhR is restoration of barrier homeostasis, a phenomenon eminent in IBD [7].

AhR is activated by metabolic compounds produced from the bacterial degradation of tryptophan [126] but also by 2,4-dihydroxyquinoline (2,4-DHQ) and quinolone derivatives [127], pyocyanin, 1-hydroxyphenazine, phenazine-1-carboxylic acid and phenazine-1-carboxamide [128]. Tryptophan degradation products such as indole, indolo [3,2-b]carbazole, indole acetic acid (IAA), 3-methylindole and tryptamine react with AhR and stimulate the production of interleukin-22 [126,129,130]. Individuals with inflammatory bowel disease (IBD) [131], metabolic syndrome [132] or celiac disease [133] have decreased fecal concentrations of AhR ligands and reduced AhR activity. Higher indole concentrations were detected in patients suffering from *Clostridioides difficile* (*Clostridium difficile*) infection (CDI) compared to healthy individuals [134]. Elevated levels of indole produced by enterotoxigenic *E. coli* stimulated the colonization of *C. difficile* whilst other intestinal bacteria were repressed [134]. When indole was administered to GF mice, the expression of genes encoding tight junction proteins increased and improved the resistance of epithelial cells to colitis [135]. Since Trp is not synthesized by gut microbiota or the host, indole levels are directly linked to diet. Roasted cashew nuts, sunflower seeds, cheddar cheese, chicken breast and boiled eggs are rich in Trp [136].

The importance of AhR cannot be overemphasized as it plays a crucial role in barrier integrity; intestinal and immune homeostasis, mostly due to the regulation of tight junction proteins; the generation and survival of intraepithelial lymphocytes (IELs); the production of IL-22 and IL-10; the regulation of peristalsis and microbiota density; the regulation of goblet cell differentiation in the colon, specifically preventing goblet cell depletion in the elderly [2,122,127,137]; and the stimulation of antimicrobial peptide production via IL-22 [138,139,140,141,142]. Individuals with IBD and celiac disease have low levels of AhR in their feces [133]. Similarly, patients with Crohn’s disease have decreased levels of AhR, especially in the ileum, and have the tendency to convert ILC3 to type 1 innate lymphoid cells (ILC1) [143,144]. ILC1 elicits the production of IFNγ and TNFα, which are both known to induce apoptosis of epithelial cells in vitro, and most likely also in vivo, disrupting the intestinal barrier and increasing microbial-driven inflammation [145]. LC1 also induces TGFβ, which is associated with tumor formation and colorectal cancer (CRC) [145]. Further research is required to have a better understanding of the cross-talk between host AhR and bacterial QS molecules.

## 5. Can QS Be Used to Control Microbial Infections?

Five years ago, the World Health Organization (WHO) published a list of pathogenic bacteria most resistant to currently used antimicrobials. *Acinetobacter baumannii*, *P. aeruginosa* and enterobacteria resistant to carbapenems and species producing extended spectrum beta-lactamases (ESBLs) were amongst the top ones on the list [146]. This urged many scientists to investigate the possibility of using anti-QS therapy, referred to as quorum quenching (QQ), to prevent or control bacterial infections [147]. In recent years, many published articles have reported promising results indicating the possibility of reducing the pathogenicity of microorganisms and easier eradication when co-treated with antibiotics. Wu et al. [148] suggested that AHL-based QS may be used to control infections caused by Gram-negative bacteria. This approach is only possible where disrupting QS has a major effect on the expression of virulence genes [149]. Limited successes were reported using QQ in the treatment of infections caused by *P. aeruginosa* and *S. aureus* [150], notably concerning biofilm-associated infections [151]. The rationale behind this concept is to interrupt receptor proteins, degrade autoinducing signals or inhibit the synthesis of QS signaling molecules [152,153,154,155]. Another approach is using synthetic compounds analogous to QS signaling molecules [156].

Van den Abbeele et al. [157] reported a decline of approximately 60% of mucosa-associated pathogens, mostly *Clostridium* spp., when QQ was applied. Although promising from a perspective of infection management, such drastic changes may lead to the development of pro-inflammatory diseases such as cystic fibrosis [158], sclerosis [159] and IBD [160,161] and an increase in *Enterococcus* and *C. difficile* cell numbers [162]. Perhaps the most alarming of all is evidence of increased cell aggregation and biofilm formation in bacteria with a dysfunctional or absent luxS QS system, as reported for *Helicobacter pylori* [163], *Vibrio cholerae* [164], *Aggregatibacter actinomycetemcomitans* [165], *Actinobacillus pleuropneumoniae* [166], *Haemophilus parasuis* [167], *S. aureus* [168], *S. epidermidis* [169], *Streptococcus mutans* [170], *Enterococcus faecalis* [171] and *Bacillus cereus* [172]. Meropenem and levofloxacin stimulated the expression of an efflux pump in *A. baumannii* that promoted the release of an AHL, resulting in an increase in QS-mediated biofilm formation [173]. Ciprofloxacin, ceftazidime and azithromycin prevented QS when used at sub-inhibitory concentrations [174]. Despite these challenges and limitations, QQ may still be the answer to controlling bacterial infections [147,153,154,155,175,176,177], especially those caused by multi-drug-resistant (MDR) and extensively resistant microorganisms (XDR) [178]. However, with the variable reactions of pathogens to QQ therapy combined with antibiotics, the treatment of infections will have to be evaluated on a case-to-case basis.

Strains may develop resistance to QS inhibitors, as shown for *P. aeruginosa* exposed to brominated furanones. Resistant cells had mutations in genes encoding efflux pumps [179]. Strains of *P. aeruginosa* resistant to carbapenems and azithromycin lost antibiotic-associated QS inhibition [180]. *Streptococcus pyogenes* and *S. aureus* mutants without LuxS (Δ*luxS*) were more resistant to macrophages [181,182]. This may lead to the development of persistent pathogens difficult to eradicate. For further information on resistance to QQ, the reader is referred to Defoirdt et al. [183], García-Contreras et al. [179], Kalia et al. [184] and Liu et al. [185].

In the case of *S. aureus*, the accessory gene regulator (agr) regulates the expression of several genes, including virulence factors such as formylated peptides, proteases and pore-forming toxins (PFTs) such as hemolysins, leucocidins and phenol-soluble modulins (psms) [38]. Strains of *S. aureus* lacking the *agr* gene (Δ*agr*) are more prone to causing chronic infections and bacteremia [38,39]. These findings suggest that treatment of *S. aureus* infections with QQ is not an option.

*Pseudomonas aeruginosa* uses psms to alter cell membrane properties [186] and activate the immune system [187]. These amphipathic peptides lyse neutrophils, erythrocytes and T cells [186]. By using QS inhibitors, the number of microbial enzymes degrading mediators of inflammatory responses may be limited. Since oxidative stress selects for cells with an active QS system [188], success in treatment using QQ techniques depends on the exposure of the infected area to oxygen and the overall immune status of the individual [189].

An interesting advance in QS research in the last decade is the discovery that QSPs may promote tumor cell invasion and angiogenesis (at least in vitro), suggesting that these peptides may stimulate stem cell differentiation and the migration of cancer stem cells [190,191]. The influence microbiota have over colon cancer stem cells, “instructing” them to become treatable or non-treatable, was raised by Trosko and Lenz [192].

## 6. Effect of QS on the CNS and Mental Health

The effect of QS molecules on the CNS is ill-researched. Several QSPs can diffuse through the intestinal mucosa and enter the circulatory system, from where they may penetrate the blood–brain barrier (BBB) [113]. Based on these findings, QSPs may play a key role in communication between the gut microbiome and the brain. If this is the case, QSPs may affect neurodevelopment and initiate neurodegenerative diseases. Further research is needed to confirm these findings.

Exotoxins produced by *S. aureus* activate the transcription factor accessory gene regulator (Agr)A, which regulates the expression of several genes, including virulence factors, pore-forming toxins (PFTs) and bacterial proteases [37]. These toxins increase intracellular calcium levels, leading to the activation of sensory neurons [193]. This is especially true for psms attached to formyl peptide receptor-like proteins (FPRs) [194]. The structural similarity of FPRs to b-defensins and ligands of mas-related G protein-coupled receptor (MRGPR) X2 [195] suggests that MRGPRs are involved in psm-mediated effects such as skin allergies [196]. The expression of FPRs in sensory and dorsal root ganglia of the colon has been well documented and linked with QS-dependent pathways involved in the gut–brain axis (GBA) [197,198]. The pore-forming toxin alpha-hemolysin (Hla) produced by *S. aureus* excites neurons by increasing the transfer of calcium [199]. According to Uhlig et al. [198], Hla produces smaller, less disruptive pores in cell membranes compared to psms [200]. The authors have also observed the expression of Adam10, a membrane-bound metalloprotease produced in sensory neurons, to which Hla binds [201]. The importance of exotoxins in GBA communication is unknown. However, since *S. aureus* is associated with irritable bowel syndrome and food [202,203], these QS molecules have the potential to directly modulate gut–brain communication and intestinal reflex.

Janssens et al. [204] screened 85 quorum sensing peptides on six different neuronal cell lines and found 22 peptides with a possible effect on the GBA. Of these, four peptides induced neurite outgrowth, two peptides inhibited nerve growth factor (NGF)-induced neurite outgrowth and eight peptides induced neurite outgrowth in human SH-SY5Y neuroblastoma cells. Two peptides killed SH-SY5Y cells and six peptides induced either IL-6 expression or nitric oxide (NO) production.

Several reports have been published on the role that cell wall components such as lipopolysaccharides, polysaccharides and peptidoglycans play in neuron activation and the GBA [37,205,206,207,208]. Cell wall components also induce the release of neuropeptides, ATP and cytokines [209]. Short-chain fatty acids, tryptophan, trace amines [142,210] and exotoxins [182] also have neuromodulator properties. Serotonin and histamine excite mast cells in the proximity of nerve endings [210,211].

Neuronal conditions such as Alzheimer’s disease (AD), autism spectrum disorder (ASD), multiple sclerosis (MS), Parkinson’s disease (PD) and amyotrophic lateral sclerosis (ALS) are associated with dysfunctional microglia [212]. Fecal transplants from humans with attention deficit hyperactivity disorder (ADHD), AD and PD to mice activated microglia in the brain and aggravated cognitive and physical impairments [213,214,215]. These findings along with more evidence of a clear link between microbial dysbiosis and neurodevelopmental, neurodegenerative and psychiatric disorders such as ASD, schizophrenia, AD, major depressive disorder (MDD) and PD [216,217,218,219] prompted researchers to have a closer look at the GBA. For more information on gut bacteria and neurotransmitters, the reader is referred to a recent review by Dicks [220]. The role that gut bacteria play in neuropsychiatric disorders has recently been reviewed by Dicks et al. [221]. 

## 7. Conclusions

With bacteria representing the largest fraction of almost 3000 microbial species in the GIT, it is no surprise that they have developed mechanisms to communicate with host cells. Some QS molecules are genus-specific, but a few are used by Gram-negative and Gram-positive bacteria. Hormones such as Epi and NE and certain carbohydrates (e.g., fucose and EA) activate specific receptors in bacteria that, in turn, trigger sensors in the cytoplasm to regulate gene expressions. In a healthy GIT, these signaling molecules are important in maintaining a homeostatic status. Some QS molecules, such as 3-oxo-C12:2-HSL, protect tight junction proteins and may be important in the treatment of leaky gut syndrome. Some QS molecules stimulate tumor growth and are closely associated with the development of specific cancers, whilst others are linked to neurological disorders. QSPs that penetrate the blood–brain barrier (BBB) constitutes an area that warrants more research, especially since the gut microbiome is increasingly recognized as a key player in neuropsychiatry. This warrants a better understanding of the underlying mechanisms involved in the regulation of gut bacterial QS systems. We also need to have a better understanding of bacterially produced QS molecules in adrenal pathways. With more in-depth knowledge of the different QS systems produced by intestinal bacteria, we may be able to develop biomarkers that can be used to diagnose neurological and psychiatric diseases such as anxiety and depression, MDD, schizophrenia, bipolar disorder, autism and OCD. More research is required to understand the integral communications between autoinducer-based signaling molecules and neurons in the human CNS. Research on using QS inhibitors (QQ therapy) to control microbial infections is gaining interest. Although a microbial infection may not be fully controlled using this strategy, it may provide the host’s immune system a better chance to overcome the infection. Before this approach can be implemented, we need to investigate the effect that QS inhibitors may have on non-pathogenic, beneficial, commensal gut microbiota.

## Figures and Tables

**Figure 1 microorganisms-10-01969-f001:**
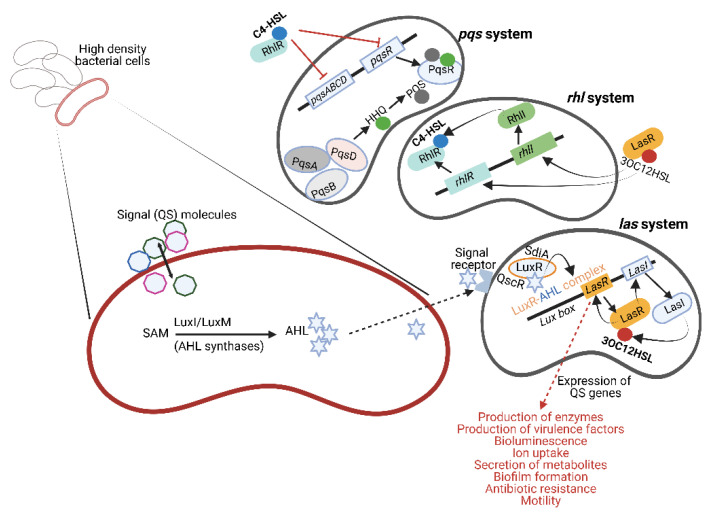
Quorum sensing (QS) molecules produced by Gram-negative bacteria. Acyl-homoserine lactone (AHL) is produced from S-adenosylmethionine (SAM) by AHL synthase (shown on the **left**). LuxI-type synthases are major producers of AHLs. LuxM synthase, described for *Vibrio harveyi*, is important in intra-species communication. Signal receptors on the surface of the responding bacterial cell recognize AHL, form a LuxR–AHL complex and induce genes in the *Lux box* to produce 3-oxo-C12-homoserine lactone (3-oxo-C12-HSL, abbreviated as 3OC12HSL in this figure to conserve space). This is referred to as the *las* system. In the absence of LuxI synthases, LuxR proteins detect different AHL molecules produced by other bacterial species. Examples of these receptors are SdiA (LuxR homolog) in *E. coli* and QscR in *P. aeruginosa*. *Pseudomonas aeruginosa* has three major QS systems, namely *las*, *rhl* and *pqs* (shown on the **right**), that mediate cell-to-cell communication and control the synthesis and secretion of virulence factors, bioluminescence, biofilm formation, etc. Additionally, 3-oxo-C12-HSL may also induce the *rhl* system to produce C4-homoserine lactone (C4-HSL). The *pqs* system uses two signal molecules, i.e., 2-heptyl-3-hydroxy-4(1H)-quinoline (also referred to as *Pseudomonas* quinolone signal or PQS) and its biosynthetic precursor 2-heptyl-4-hydroxyquinoline (HHQ). This illustration was constructed using BioRender (https://biorender.com/, assessed on 27 September 2022).

**Figure 2 microorganisms-10-01969-f002:**
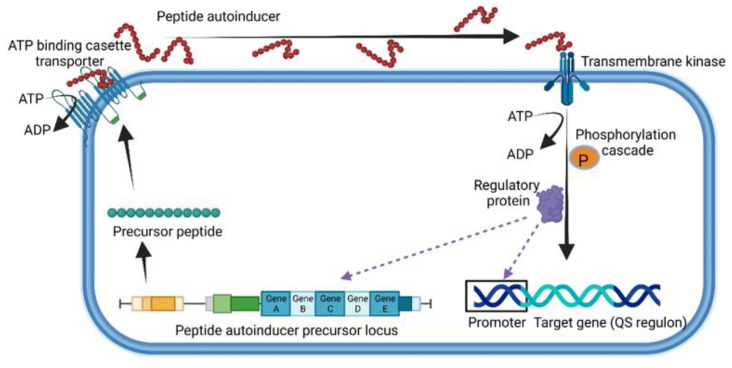
General representation of quorum sensing used by Gram-positive bacteria. This illustration was constructed using BioRender (https://biorender.com/, assessed on 1 September 2022).

**Figure 3 microorganisms-10-01969-f003:**
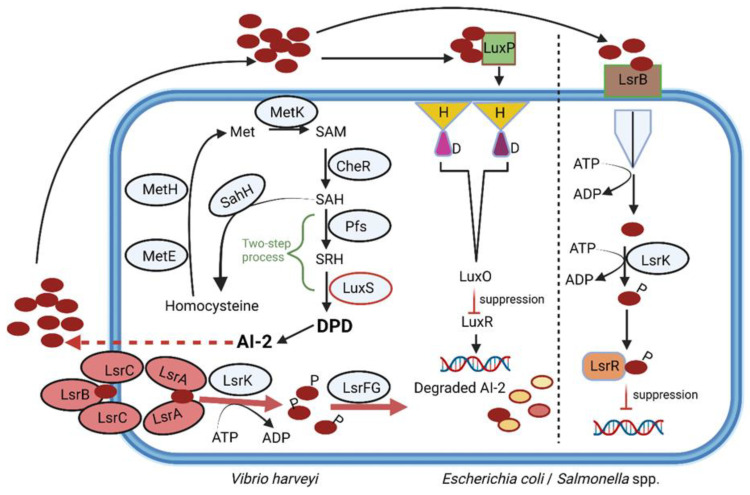
Production of AI-2, regulated by LuxS (an autoinducer synthase). S-adenosylhomocysteine (SAH) is converted to homocysteine by SAH hydrolase (SahH) in a one-step reaction but may also be produced from the cleavage of the thioether linkage of S-ribosylhomocysteine (SRH). This is a two-step reaction that requires SAH nucleosidase (Pfs) and LuxS. The intermediate 4,5 dihydroxy-2,3-pentanedione (DPD) is rearranged to form AI-2. Red circles = AI-2; LsrB-LsrC-LsrA = Lsr ABC-type transporter; LsrK = Lsr kinase; LsrFG = genes F and G, part of the Lsr operon; MetE = cobalamin-independent methionine synthetase; MetH = cobalamin-dependent methionine synthetase; MetK = adenosylmethionine synthetase; CheR = methyltransferase; P = phosphate group; LuxO = a central regulator; LuxR = repressor. This illustration was constructed using BioRender (https://biorender.com/, assessed on 5 September 2022).

**Figure 4 microorganisms-10-01969-f004:**
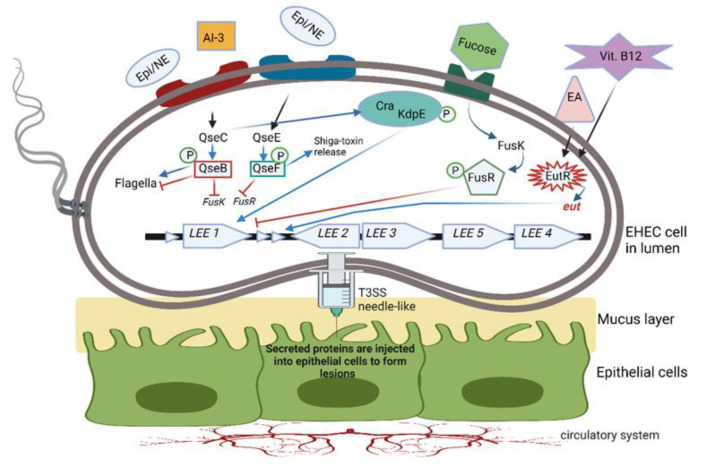
Two-component QS system (TCS) used by enterohemorrhagic *E. coli* (EHEC). Bacterial cells sensing environmental signaling compounds such as epinephrine (Epi), norepinephrine (NE), autoinducer 3 (AI-3), fucose, ethanolamine (EA) and vitamin B12 activate transmembrane histidine kinase receptors (indicated here as blue and red rectangles). Response regulators either activate or repress the TCS. QseC histidine sensor activates QseB, and QseE activates QseF. QseB regulates the expression of flagella and, at the same time, represses the expression of *fusK/-R*, involved in fucose metabolism, and the expression of virulence genes. QseC controls the response regulator KdpE, which, together with Cra, stimulates genes in the LEE operon to form microscopic T3SS needle-like structures through which proteins are injected into the host cell. Microvilli on the surface of epithelial cells are eradicated, and lesions with actin-rich pedestals form, onto which EHEC cells attach. QseC also activates regulator QseF, which stimulates the production of Shiga toxin. FusK is activated by fucose and phosphorylates FusR to inhibit LEE expression. Genes in the *eut* operon encode the EutR transcription factor, which recognizes the presence of ethanolamine (EA) and vitamin B_12_. Both compounds are required to promote the transcription of genes in the LEE operon. This illustration was constructed using BioRender (https://biorender.com/, assessed on 3 September 2022).

**Figure 5 microorganisms-10-01969-f005:**
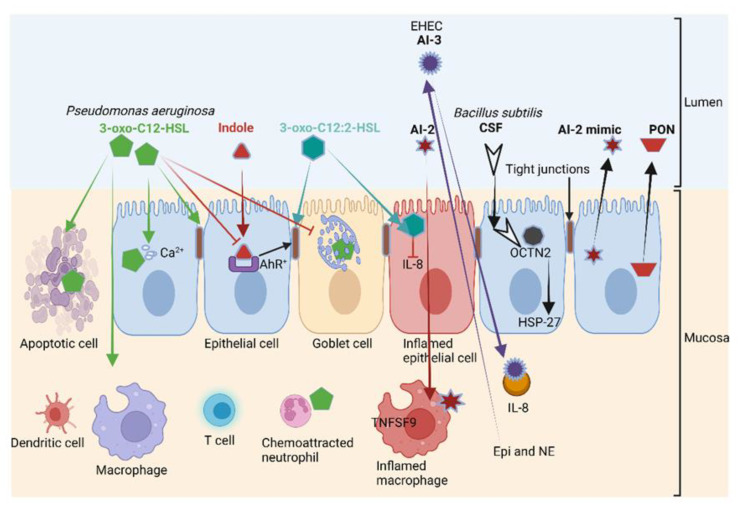
The 3-oxo-C12-HSL produced by *P. aeruginosa* (also shown in Figure 1) induces apoptosis in various cell types, including epithelial cells; disrupts tight junctions and decreases mucin production. Intestinal acyl-homoserine lactone (3-oxo-C12:2-HSL) and indole (produced from tryptophan) protect tight junctions. The competence and sporulation factor (CSF), a pentapeptide produced by *B. subtilis*, binds to the cation transporter OCTN2 and activates heat shock protein 27 (HSP-27), which increases the strength of intestinal barriers. Meanwhile, 3-oxo-C12-HSL stimulates chemoattraction and phagocytosis in neutrophils and induces cell death. The autoinducers AI-2 and AI-3 induce the expression of the immune mediators TNSF9 and interleukin (IL)-8, respectively, leading to the inflammation of macrophages, while 3-oxo-C12:2-HSL reduces IL-8 production. Epinephrine (Epi) and norepinephrine (NE) bind to the AI-3 receptor in enterohemorrhagic *E. coli* (EHEC), thereby interfering with QS. Intestinal epithelial cells secrete a mimic form of AI-2 and paraoxonase (PON) to degrade HSLs. AhR: Aryl hydrocarbon receptor. This illustration was constructed using BioRender (https://biorender.com/, assessed on 6 September 2022).

**Figure 6 microorganisms-10-01969-f006:**
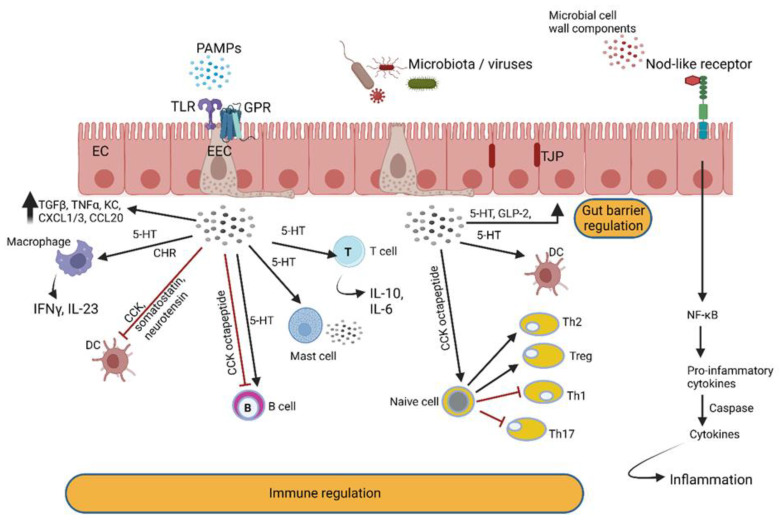
Enteroendocrine cells (EECs) in the gut wall detect intestinal bacteria and microbial metabolites and react by secreting peptide hormones and cytokines that react with immune cells. Hormones produced by EECs modulate intestinal barrier functions and react with enteric nerves. The latter communicate with the central nervous system via the vagus nerve. PAMPs = pathogen-associated molecular patterns; TLR = Toll-like receptor; GPR = G protein-coupled receptor; TGF = tumor growth factor; TNF = tumor necrosis factor; KC = neutrophil chemokine; CXLC = chemokine (C-X-C motif) ligand; CCL = chemokine ligand; IFN = interferon; IL = interleukin; CCK = cholecystokinin; DC = dendritic cell; 5-HT = serotonin; Th = T helper cell; Treg = regulatory T cell. This illustration was constructed using BioRender (https://biorender.com/, assessed on 6 September 2022).

**Figure 7 microorganisms-10-01969-f007:**
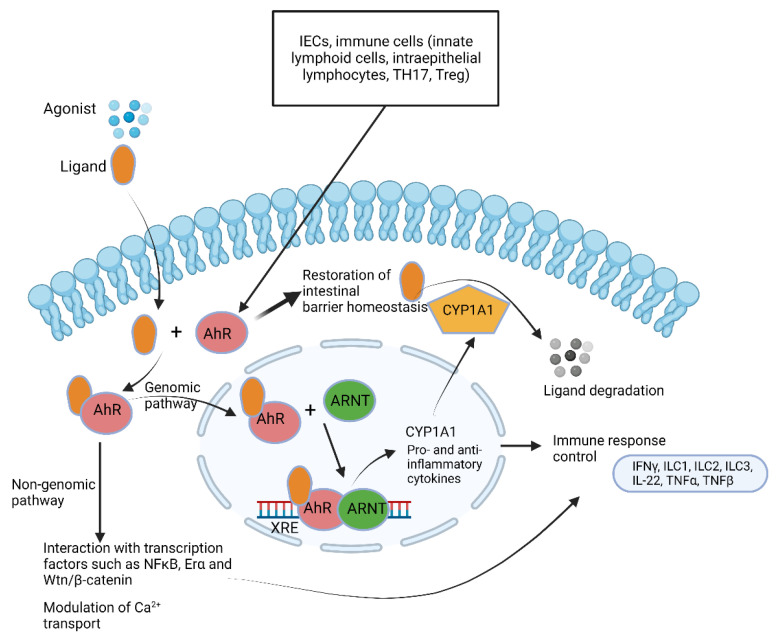
Summary of aryl hydrocarbon receptor (AhR) pathways. AhR ligands such as 6-formylindolo [3,2-b]carbazole; 2,3,7,8-tetrachlorodibenzo-p-dioxin and polycyclic aromatic hydrocarbons cross the cytoplasmic membrane and bind to AhR. The ligand–AhR complex crosses into the nucleus and forms a heterodimer with the AhR nuclear translocator (ARNT). Binding of the ligand–AhR–ARNT complex with xenobiotic response elements (XREs) leads to the expression of cytochrome P450, family 1, subfamily A, polypeptide 1 (CYP1A1). CYP1A1 degrades AhR ligands. This leads to the induction of several genes encoding pro-inflammatory and anti-inflammatory cytokines. The non-genomic pathway involves interaction between AhR and other transcription factors and modulates Ca^2+^ transport. This illustration was constructed using BioRender (https://biorender.com/, assessed on 6 September 2022).

## Data Availability

Not applicable.

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
