# Peer review of "How does Quorum Sensing of Intestinal Bacteria Affect Our Health and Mental Status?"

_microorganisms, 2022, doi:10.3390/microorganisms10101969_

Round 1

Reviewer 1 Report

The Author made an in depth overview about what it is known and the importance of QS of selected probiotic strains to be used in the near future as essential biomarkers in the diagnosis of several diseases such as IBD, IBS, and CRC.

Minor changes:

Genus and bacterial species should be always in italics, with the genus initials always capitalized. Please check all the text

Author Response

Dear Reviewer

All species names have been checked.  Serovars, e.g. Salmonella enterica serotype Typhimurium and S. enterica serotype Typhi are not in italics and correct as in the manuscript.

Yours sincerely

Prof LMT Dicks

Reviewer 2 Report

Role of quorum sensing (QS) in gut is a very relevant topic in the study of exploration to clarify the relationship within gut microbiota or between gut microbiota and host. The effects of QS signals on host intestinal cells and sensory neurons were also introduced and speculated in detail. And diagrammatic representation of the role of quorum sensing is well depicted.

1) Line 10: Is there an extra space before "Although" in the manuscript? and you need to check the manuscript.

2) Lines 12-17: QS signals are fully covered in the abstract but are not necessary. It should be shortened. You can describe the relevant content in detail in the introduction.

3) Lines 52-83: The interbacterial communication of Gram-negative bacteria is not well summarized. The two paragraphs are not well related, nor are they related to the theme of the manuscript. More enteric Gram-negative bacteria should be listed for QS. Such as Escherichia coli and Salmonella. I suggest reorganizing this section.

4) Lines 163-165: "Changes in populations of Firmicutes and Bacteroidetes alter the level and composition of SCFAs, that, in turn, affect gene expressions, cytokine secretion and regulatory T cell induction. "Please add references to this section.

5) Line 680: Please clear the underline below the author, and you need to check the manuscript.

Author Response

Role of quorum sensing (QS) in gut is a very relevant topic in the study of exploration to clarify the relationship within gut microbiota or between gut microbiota and host. The effects of QS signals on host intestinal cells and sensory neurons were also introduced and speculated in detail. And diagrammatic representation of the role of quorum sensing is well depicted.

  • Line 10: Is there an extra space before "Although" in the manuscript? and you need to check the manuscript.

Answer:

All sentences in the manuscript are separated by two spaces and was left as is.

  • Lines 12-17: QS signals are fully covered in the abstract but are not necessary. It should be shortened. You can describe the relevant content in detail in the introduction.

Answer:

The abstract has been shortened by not mentioning the signaling molecules by name (see lines 11-13).  The information has now been included in the Introduction section (see lines 30-37).

3) Lines 52-83: The interbacterial communication of Gram-negative bacteria is not well summarized. The two paragraphs are not well related, nor are they related to the theme of the manuscript. More enteric Gram-negative bacteria should be listed for QS. Such as Escherichia coli and Salmonella. I suggest reorganizing this section.

Answer:

The section on Gram-negative bacteria (section 2.1) has been reorganized and additional information has been added pertaining to E. coli and Salmonella (lines 68-87).  The caption of Figure 1 has been rearranged (lines 117-127) and additional information (SdiA and QscR) has been added in the drawing.

4) Lines 163-165: "Changes in populations of Firmicutes and Bacteroidetes alter the level and composition of SCFAs, that, in turn, affect gene expressions, cytokine secretion and regulatory T cell induction. "Please add references to this section.

Answer:

A reference has been inserted (line 191) and added to the reference list.

5) Line 680: Please clear the underline below the author, and you need to check the manuscript.

Answer:

The underlining has been deleted (line 772, reference 80).  The text referencing the paper has been amended (line 231).

Reviewer 3 Report

This is a review of quorum sensing, summarizing research well known to those in the field nevertheess it can be useful for lay users. The novelty of the conclusions could be enhanced. 

Author Response

This is a review of quorum sensing, summarizing research well known to those in the field nevertheess it can be useful for lay users. The novelty of the conclusions could be enhanced

Answer:

The conclusion section has been amended.

Reviewer 4 Report

The present review by Leon M. T. Dicks tries to encompass the gathered knowledge on one of the most studied topics for bacteria signaling and adaptation, Quorum sensing. Several reviews on this topic have been published over the years to simplify the state of the art and serve as a very useful source to dissect a topic with vast inputs from different research groups. 

Recent reviews such as those from Abisado et al., 2018 (mBio), Sampriti Mukherjee & Bonnie L. Bassler (Nat reviews, 2019) and Liang Wu and Yubin Luo (Frontiers in Microbiology, 2021) also present information which is very similar to the one presented by Prof. Dicks. Due to this reason, I do not recommend this work to be published in its current form without a revision, this a strong negative point, the author makes too much emphasis on repeating the information, which has been presented so many times before. The basis of quorum sensing have been reviewed repeatedly and there is no new information brought forth here by the author on this regard. I would recommend to reduce the introductory section (everyone can always refers to so many previous works and reviews on the properties of autoinducers and quorum sensing in general) and expand on the final sections, where the author discuss the health impact and interaction with the immune system and host factors, here lies the novelty of this manuscript. Here, I found myself very interested and intrigued by the information provided, the information given here is well structured, easy to assimilate and very well supported by the figures.

Even though the work in itself is not novel on the content (too much on the quorum sensing basics) the author shows a careful and well-made effort to bring in and discuss relevant literature to support his line of discussion regarding intestinal bacteria quorum sensing and our overall health. The figures are very well made and simple enough to understand the compendium of information gathered and presented here by the author. The choice of references is good and they are properly discussed.

I would recommend that the author reviews the introduction, reducing the length of sections 1 and 2 and expand on the discussion for sections 5 and 6.

Author Response

Answer:

Thank you for the valuable points made.  Sections 5 (Can QS be Used to Control Microbial Infections?) and 6 (Effect of QS on the CNS and Mental Health) have been expanded.  I do feel that a broad understanding of QS is needed and the text preceding sections 5 and 6 was kept as is, except for changes recommended by the three other reviewers.  One of the reviewers requested additional information on Gram-negative QS (sections 2.1), which was added.

Round 2

Reviewer 4 Report

Even though I believe this review could have used a further expansion of the last 2 sections on therapy and effects on the mental health to increase its impact, the author has prepared a very thorough and complete review of QS. The review is enjoyable to read, the figures are engaging and very illustrative and the narrative is very well estructured, I enjoyed reading this review very much.